# Challenges in Transition of Care for People with Variations in Sex Characteristics in the European Context

**DOI:** 10.3390/healthcare12030354

**Published:** 2024-01-30

**Authors:** Martin Gramc

**Affiliations:** Institute of Biomedical Ethics and the History of Medicine, University of Zürich, 8006 Zürich, Switzerland; martin.gramc@ibme.uzh.ch

**Keywords:** transition of care, resources, hurdles, psychosocial support, variations in sex characteristics

## Abstract

Objective: People with variations in sex characteristics (VSCs) have been receiving inadequate care for many decades. The Chicago consensus statement in 2006 aimed to introduce improved comprehensive care, which would include the transition of care from pediatric to adult services organized by multidisciplinary teams. Yet, the evidence for transitional care is scarce. The aim of this paper is to outline the delivery of transition of care for adolescents and young adults with VSCs. Method: Seven focus groups were conducted with health care professionals and peer support groups by care teams in Central, Northern, and Western Europe. The data from the focus groups were examined using reflexive thematic analysis. Results: Even though the transition of care has been implemented in the last two decades, it remains inadequate. There are differences among countries, as the quality of care depends on available resources and variations in sex characteristics. Moreover, there are significant hurdles to adequate transition of care, as there is lack of time and funding. The lack of adult care providers and psychosocial support often leaves young adults with VSCs to navigate the health care system alone. Conclusion: The outcome of the study shows that the transition of care is organized through the department of pediatric endocrinology. The quality of care varies due to resources and variations in sex characteristics. A lack of adult specialists, and especially psychosocial support, represents the biggest obstacle for young adults and adults in navigating the health care system and for improvements in the provision of health care to adults. There is a risk of re-traumatization, as adolescents and young adults must often repeat their medical history and educate adult care providers who are insufficiently trained and knowledgeable.

## 1. Introduction

Variations in sex characteristics (VSCs) refer to bodily attributes that cannot be classified as typically male or female [1]. As such, they may require collaborative care from multiple care providers [2]. Collaborative care is a multiprofessional approach to patient care based on clinical evidence and standardized tools to provide care tailored for an individual and is accompanied by follow-ups and communication among health care professionals [3]. In 2006, the Chicago consensus statement was introduced to improve the quality of care for people with VSCs and their parents [4]. Quality of care consists of effective, safe, and patient-centered care as well as accessibility, efficiency, and equity of care [5,6]. The consensus advocated for long-term multidisciplinary care, psychosocial support, complete disclosure of medical diagnosis and treatment, and caution against early cosmetic surgery, and introduced new terminology [4,7]. It indicated that multidisciplinary teams should educate other health care professionals involved in the care for people with VSC and their parents and develop a plan for care that should be patient-centered [4,7,8,9]. Yet, the consensus statement did not result in a complete change in medical practice, as the evidence suggests that surgeries continued [10,11]. The provision of multidisciplinary care has not been adequately implemented [12,13], and the terminology was met with critique by the intersex movement and some medical professionals [14,15,16].

The consensus statement acknowledges the historic deficiency in the transition of care [17], briefly mentioning that one of the responsibilities of multidisciplinary teams is coordinating the transition of care from pediatric to adult settings [4]. The consensus update added that people with VSCs share challenges in accessing adult care with other people with chronic conditions [9]. Moreover, the consensus update also highlighted the reasons for the lack of appropriate transition of care: Adolescents and young adults avoid care providers due to poor outcomes of past medical treatments, non-disclosure of diagnosis and medical treatment in the past, and anxiety related to sexual and romantic relations [9]. In the research on the adjustment from pediatric to adult care, transition of care is defined as a process that includes services aimed at planned and coordinated shifts between pediatric and adult care for young people with chronic conditions [18,19,20].

The few recommendations on transition of care for adolescents and young adults with VSCs that were published in the years between the consensus and its update in 2016 stipulated that the transition of care is a multiple-year process that requires a plan and review for each person with VSCs according to their variation and needs, an assessment of readiness for the transition, and assistance in guiding and educating a person with VSCs and their family [17,21,22,23]. Furthermore, the recommendations highlighted the importance of continuous collaboration between pediatric multidisciplinary team, parents, peer support groups, and adult providers who should be involved in the care as early as possible, as well as that adult providers should have good communication skills and be well-versed in issues surrounding care for people with VSCs. These recommendations also pointed out that transition is a process that involves a shift in the care approach, and it should start at the age of 12–13 years and be finished around the age of 25, whilst the transfer is a step in which a patient stops seeing a pediatric health care professional and makes an appointment with an adult care provider [17,22].

Although there is increasing awareness among professionals regarding the requirements for transitioning care, the evidence indicates that there is still significant opportunity for enhancement [20]. Adults with VSCs experience health problems that have been either created or worsened by medical interventions [24]. The research has identified several obstacles to the successful transition of care, for example, insufficient training in adolescent health care, potential diffusion of responsibility among pediatric and adult care providers, the complexity of health care conditions, inadequate communication between pediatric and adult health care providers, the need for meaningful inclusion of adolescents and young adults in the transition process, lack of planning and coordination of services, and insufficient evidence to guide the development of transition of care [20,25,26,27].

Only a limited number of studies have investigated the difficulties of transitioning care for adolescents and young adults with VSC [24]. These studies have identified problems such as: a lack of providers trained in adult care for people with VSC and their parents, a lack of priority and finances for multidisciplinary care in healthcare systems that VSCs require, differences in care approaches between pediatric and adult care, a lack of psychosocial support, and a lack of data on the provision of transition care [21,22,28]. The scarce evidence on the transition of care points to a need to examine the poor implementation of transitional care [28,29].

This study utilized an exploratory design and approach based on reflexive thematic analysis [30,31] that would provide the author with a more reflexive and subjective outlook with which to outline the scope of issues in the field. The author of the present study is part of a broader research project called INIA—Intersex New Interdisciplinary Approaches—that aims to develop knowledge that will inform policy making and practice that supports the wellbeing and social and economic contributions of people with variations in sex characteristics. Participation in the project has enabled the author to become familiar with the issues in care for people with VSCs and the relevant stakeholders. The research question was addressed through constructionist, experiential, inductive, critical, and latent frameworks. The aim of the study is to map out the delivery of transition of care for adolescents and young adults with VSCs based on focus groups conducted with members of peer support groups and multidisciplinary teams working in care for children with VSCs.

## 2. Materials and Methods

### 2.1. Design

The study used an exploratory qualitative design based on mixed focus groups (in terms of professional background) and reflexive thematic analysis [30]. Reflexive thematic analysis was used because the author wanted to map out the meaning patterns and divergences in viewpoints of the participants regarding the approaches to care. The reflexive thematic analysis approach [31] was flexible enough to enable the author to use their creative and critical thinking skills to draw out the underlying themes from the focus groups. The study design was informed by the concepts of pediatric shared decision making [32], multi/interdisciplinarity [33], and children’s rights in health care [34]. In semi-structured focus groups, an interview guide was used to obtain information about the participants’ viewpoints on the composition, collaboration, and challenges between peer support groups and members of multidisciplinary teams working in care for children with VSCs and their parents.

### 2.2. Selection of Participants

The inclusion criteria for the participation in the study were as follows. Participants had to be health care professionals working in a European multidisciplinary team providing care for children with VSCs and their parents, or members of peer support groups involved in any way in collaboration with multidisciplinary teams. Health care professionals had to be medical specialists (primarily endocrinologists and urologists) or psychosocial support providers (psychologists and psychiatrists); and it was preferred that health care professionals and psychosocial support providers specialized in care for children with VSCs.

A purposive selection procedure was used to select participants in the field of medical care for people with VSCs on a European level. Four of the team coordinators of all the focus groups were recruited during an in-person meeting at the 9th International DSD Symposium, where the author of the study participated in a workshop facilitated by their first supervisor, who is an academic and physician in the field. The second supervisor is only an academic. The team coordinators then took the initiative to contact other team members. Efforts were made to recruit participants from different countries. Two team leaders and the team members, as well the members of the peer support groups, were selected through convenience sampling, separately from the 9th DSD Symposium. Two members of peer support group were recruited by the team coordinators who were recruited at the 9th DSD Symposium.

Ethics approval was obtained in February 2022 by the CEBES Review Board; the ethics committee of the Institute of Bioethics; and the History of Medicine, University of Zurich.

### 2.3. Data Collection and Data Setting

Six focus groups in five different European countries (Belgium, Germany, Slovenia, Sweden, and the United Kingdom) were conducted from May 2022 to February 2023 with members of multidisciplinary teams. The questionnaire guide (see Appendix A) for the focus groups was piloted with a multidisciplinary team. The pilot was conducted with a team from Switzerland in May 2022. The pilot helped the author to shorten the interview guide and to be more assertive in managing the time and group dynamics during the focus groups. All the focus groups were conducted by the author, who is a research fellow with background in sociology and gender studies and three years of experience in qualitative research. The author drafted the first version of the interview guide based on a scoping review [35] and additional literature. Then, the author discussed the interview guide with supervisors (none of them was a person with VSCs) and two PhD candidates on the INIA project, both people with VSCs. The variety of professional backgrounds and characteristics in the discussions helped the author to approach the research with respect for people with VSCs while staying critical and attuned to the complexity of the care for people with VSCs. The collaboration with supervisors and two fellow INIA PhD candidates enabled the author to enlarge the scope of interview guide, deepen the analysis of the data, and enrich the development of themes. Thereafter, the author added quotes, as suggested by the supervisors, to be used in case the participants struggled to engage in conversation. After the second round of consultation with supervisors and the fellow PhD candidates, the author changed the wording in some places. The data on transition of care presented in this paper were taken from the corpus of data collected from the focus groups, which was too comprehensive to be analyzed in one paper.

The participants were sent the information sheets and informed consent sheets a week before the scheduled time slot for the focus group, and were asked to send them signed and filled out to the author. Focus groups lasted from 45 min to 75 min, and all of them were conducted online on Zoom and were audio recorded. All focus groups were conducted on Zoom. On average, the number of participants in the focus groups ranged from 3 to 4. At the beginning of each session, the author would explain to the participants the goals of the focus groups and present the rules of the discussion. Then, the participants would introduce themselves and the reasons why they chose to participate. This would be followed by asking the questions and receiving responses. If there were any unclear answers, the author would ask for clarification or state a follow-up question. At the end of each focus group, the author would give the participants the chance to ask or state questions or concerns that arose during the discussion, but might not have been addressed. At the end of each session, the author would then thank the participants for their collaboration and ask them to send the signed informed consent sheet in case they had not done so.

The focus groups were conducted after a pilot study that included a focus group with one of the multidisciplinary teams contacted by the first supervisor and a member of a peer support group. After the pilot study, the author reduced the number of follow-up questions in the semi-structured interview guide and reduced the number of the main questions from 8 to 7 to make the focus groups less time-consuming and provide less redundant answers.

The sample consisted of peer support groups and health care professionals. The final number of participants was 18, among them 16 health care professionals and 2 peer support members. On average, the number of participants in the focus groups ranged from 3 to 4. Not all participants were included in the final selection, as 5 respondents could not attend, 2 explicitly decided not to, 2 did not provide informed consent, and 1 decided to leave the focus group at the beginning.

The focus group sessions were then transcribed and pseudonymized. The transcripts were not sent to the participants due to time constraints. The transcripts were coded with the MAXQDA 2020 program using the (following) coding tree (Table 1). The data were first independently assessed by the author, then by the second supervisor. Afterwards, the author and the second supervisor were joined by the first supervisor to compare and reflect on the assessment, which resulted in the creation of the first draft of the coding tree. The coding tree was slightly adapted after the second reading of the transcripts, as the authors made notes on already-provisionary themes and subthemes, but stayed open to changes. The coding tree was designed inductively by the author by extensive reading of the transcripts multiple times and using the notes made while conducting the focus groups. The codes were then qualitatively analyzed using reflexive thematic analysis. As the author became familiar with the data, he generated the themes, as meaning is created in the analysis process [30]. Firstly, the author focused on the commonalities in the participants’ reflections. Then, the author consulted and reflected with the research team on the emerging themes. Secondly, in the next stage, the author looked for similarities, differences, and commonalities in the transcribed material to crystalize the three most prominent themes: the organization of the transition of care, documentation, and barriers to transition. These themes featured most prominently in all the focus groups and were thus selected in the analysis. Themes like group dynamic and focus on biomedical knowledge were found only in some groups, and the author could not find a common thread in all the focus groups. The author then read the material again to see how the themes were expressed by the participants. The themes were then illustrated using selection of quotations, which were slightly revised to improve readability. The quotes from focus groups conducted in German were translated by the author into English.

## 3. Results

The three key themes and the subsequent subthemes from the focus group data concerned the organization of the transition of care (e.g., the criteria for transition, changes in the practice), documentation, and barriers to transition (lack of resources and psychosocial support).

### 3.1. Organization for the Transition of Care

#### 3.1.1. Availability of Transition of Care

Transition of care is available to some extent in all the teams, but the quality and provision vary greatly. The medical specialists in the pediatric multidisciplinary teams involved in transition of care are mainly endocrinologists, urologists, gynecologists, and psychologists, but in adult care, these medical specialties are lacking. The transition process is mostly organized within the department of endocrinology, including the variations which are not endocrinological, and it is often pediatric endocrinologists who inform and involve adult endocrinologists to participate in the team discussions when a child with VSCs is born. As participants reported the transition process for adolescents, young adults, and parents is initiated early on. The majority of participants stated that the transition of care has become well incorporated into team tasks, for example:


*We are connected also with adult endocrinologists and they are always invited also to our multidisciplinary team. So sometimes this can be of course, we can discuss these patients that are at the time of transition directly with them at the multidisciplinary team and present them to them. Otherwise, we have quite good this transition practice. So that every patient that goes to the adult endocrinologist, this is just general, not just for the DSD patients, but also other rare endocrine disorders that need further follow-ups. So that we make overview for the patient, final overview of his care under ours. And all this is then controlled when they are transferred to adult endocrinology, so that we get an information that the patient came there or something like that. So, the transition was smooth. Sometimes we even joined the patient at the first meeting. *
(Endocrinologist, FG 1)

#### 3.1.2. Criteria for Transition

The criteria for transition are multiple, and they differ among the teams. In all focus groups, the respondents pointed out that transition depends on the variation. The respondents mentioned that adolescents and young adults with VSCs should be equipped with knowledge about their variation and how to navigate the health care system. Age itself was not always mentioned as one of the criteria for transition, because in some teams, psychological maturity of the individual can be taken as one of the sufficient criteria for transition. In some teams, a special hour was devoted to talking about the transition to adult care with the individual and their parents in collaboration with the adult gynecologist or endocrinologist.


*So, yeah. Sometimes it’s age, but sometimes it’s actually maturity. So, it’s the physical mental maturity of the patient who you’d say, well, actually, I think this boy is ready, or this girl is now ready to be moving on to my colleagues. But we don’t do it as like a one-off thing. It may well be that the patient comes to that joint clinic on a few occasions for them to be getting prepared to move on. So, they may, perhaps the gynecologist or the psychologist may see them when they’re 12,13 at that clinic with the pediatric person, and then as they get older, the people will get to take over the case. *
(Endocrinologist, FG 6) 

#### 3.1.3. Changes in the Practice

There has been an improvement in the services provided, as there are centers and specialists that provide care for adolescent and young adults with VSCs. In the last two decades, the teams have managed to introduce structurally organized transition of care by investing time and resources. Most respondents have stated that the transition of care has improved, as there are now specialists for adult care and patients are referred to them.


*Over the last three years the structures have been well built which then makes everyday work much easier, because we know okay, that goes here, that goes there, simply say or write an email and then the care is definitely well provided and we can also support young people right away and don’t have to guess or puzzle. *
(Endocrinologist, FG 2) 

#### 3.1.4. Documentation

One important part of transition of care that was highlighted time and again among most participants was the documentation of the provided care and its transfer from pediatric to adult care. Participants from Germany and Slovenia mentioned that children with VSCs and their parents in most teams receive copies of the diagnosis and provided treatment, and they are also given a special folder in which they can note their questions, remarks, and impressions during consultations and collaborations with the health care providers in the teams and peer support groups.


*We then hand over all the results that we have to the adult endocrinologists, they have them in their files, and then we also make sure to schedule the appointment so that they hand over all their important documents to the young women again, because that’s often the case in childhood. The parents got the letters and then maybe they no longer have them. But we make sure that the young women then have all their documents again, the chromosome analysis, the important ultrasound examinations, that they get everything in their hands and then have them with them. So that already exists for DSD diagnoses, for Turner syndrome, for CAHs and we are still in the process of establishing that for Kleinefelter syndrome, so that we can then also carry it out in a structured way within a transition consultation. *
(Endocrinologist, FG 3) 

### 3.2. Barriers for Implementation

#### 3.2.1. Lack of Resources

The biggest obstacle to good-quality of transition of care, which was mentioned in all the focus groups, was lack of related adult specialists. Adult specialists for VSCs are not available in every hospital, which becomes a problem for adult people with VSCs who move to another town. Adult people with VSCs then either stay in touch with the pediatric specialists or are left without a specialist and seek them for a long period of time, as was pointed out by many participants.


*Although the number of specialists, of adult specialists interested in Turner DSD-pathology is rare, very rare. Adult specialists rarely have interest in these conditions, there are so many other pathologies that are much more frequent, that you may be grateful if you find somebody who is interested and has some experience and knows how to take care of adult patients. *
(a member of a peer support group, FG 5) 

Other hurdles to transition of care included poor transfer of knowledge between pediatric and adult care providers, which on the one hand forces adolescents and young adults with VSCs to educate the care providers themselves. On the other hand, pediatric care providers strive to fill in the gaps in knowledge of adult care providers, which is an extremely rare practice. Furthermore, there is a lack of financial resources for adult care, which was mentioned in relation to access MDTs, which are usually available only in university health care hospitals. A lack of local services can be very problematic, as the following quote indicates:


*In the DSD field, we try to make contacts there whenever possible. But now another young person has showed up and they moved to another town. It’s also a phase of life when people then leave their surroundings. And this young person called me from [town] saying that they don’t know anyone there, I almost said I was sorry. But that means that I can—only help him very indirectly and cannot really find anything on the spot.*
 (Endocrinologist, FG 2)

#### 3.2.2. Lack of Psychosocial Support

Another obstacle that stood out in relation to the transition of care was psychosocial support, which is severely lacking in almost all adult teams, because there are very few psychosocial providers. There is a lack of knowledge of adult psychosocial care among health care professionals, which leads to a lack of empowerment for adolescents and young adults with VSCs, as was mentioned by some respondents:


*There is no program, but organizationally I’m in the medical psychology unit and adult patients, when they transition to adult care, they can get referred to medical psychology, which has a clinic of its own… I have some adult patients that speak about not feeling empowered enough because they are still trying to grasp the diagnosis themselves, and once they’re through that stage, they don’t feel ready for the leaving groups.*
(psychologist, FG 4) 

Overall, the key themes from this study concern resources and psychosocial support. The multidisciplinary teams have managed to organize the transition of care as part of their tasks, but not adequately, as the transition of care still lacks in terms of the provision of psychosocial support and adult specialists, especially those patients whose variations are not addressed by endocrinologists.

## 4. Discussion

The findings of this study show that transition of care has been unevenly implemented in the last two decades, although there are teams that have organized transition of care. A minority of teams have introduced an early transition process, as pediatric health care professionals feel responsible for educating adult health care providers. Furthermore, these few pediatric teams have even introduced an innovative practice in documentation, as they provide special folders for parents to use to keep track of health care delivery and discussions with health care professionals. Special folders seem to be unique to the care for people with VSCs, as in other areas of health care, handheld notes are used.

However, the lack of resources and psychosocial support represent the most serious hurdles to proper transition of care that came out of the focus groups, and the findings suggest that transition of care has been inadequately implemented. The results of the study suggest that the quality of transition of care is not effective, safe, or patient-centered, as care services suffer from a lack of adequately skilled health care professionals and poor outreach to the adolescents and young adults with VSCs. The lack of resources found in this study has already been indicated in previous studies [29,36]. There are no continual examinations, centers of care are remote, and practitioners in adult care lack knowledge on variations in sex characteristics [29,36,37]. The findings of this study are consistent with general studies on transition of care in adolescents and young adults with chronic conditions, which have indicated insufficient coordination of services and lack of recognition of who should take responsibility for the transition [25,26].

Our study went further and showed that transition of care often cannot happen, as there are often simply no adult care providers for adolescents, young adults, or older people with VSCs. The lack of adult providers for adolescents and young adults with VSCs corresponds to the lack of access to health care professionals in Europe [38]. For example, in Germany, doctors are unequally represented throughout the regions, whereas in the United Kingdom, not only are doctors more concentrated in some regions, but there is an overall lack of health care professionals. The uneven representation of health care professionals is further complicated by the fact that medical specialties like internal medicine, psychiatry, pediatrics, pathology, and neurology receive more funding than family medicine, gynecology, and plastic surgery [39]. Additionally, variations in sex characteristics are seen as a non-viable option for the future by some health care professionals and parents [40]. The growing child is envisioned to dispense with options and fit into the “cis-gendered future” [16]. The discourse on the cis-gendered future informs the medical practice, as it signals that adult people with VSCs had their variations removed or made invisible, and that care for it is no longer needed.

Furthermore, even if there are adult care providers, this does not guarantee that the care will be provided as they lack knowledge and expertise, which means that adolescents, young adults, and older people with VSCs must educate their adult care providers to ensure that care is provided in the first place. This represents an emotional burden for the adolescents, young adults, older people, and engaged parents and families who try to support the young person into adulthood, as they must often repeatedly share their medical history with different care providers. The need to repeat one’s medical history to inadequately educated and trained adult professionals may re-traumatize the adolescents and adult people with VSCs. This is consistent with previous studies that showed how the burden on adults with VSC to educate their adult practitioners causes feelings of frustration and stress to develop [36,37]. But it is, sadly, not uncommon in the transition process; research in the past has highlighted that medical professionals receive minimal training on adolescent health care and transition of care, and that the transition process is not centered around adolescents’ needs [20,27].

This is consistent with findings from the study by Sanders and Carter [41], in which young women with VSCs reported inadequate support to address their emotions. Even though, in some teams, the communication between pediatric and adult care providers is established from the start, there is still room for improvement to prevent repeated sharing of medical history on the patient’s behalf [42]. The practitioners could avoid major barriers in overall transition of care to ensure suitable provision by employing more adult care practitioners; establishing a relationship between adult providers, adolescents, and young adults; and developing skills in adolescents/adults to navigate the health care system, which is quite challenging for people with rare conditions such as VSCs, as is stated in the literature [43].

The results of this study confirm the differences in care approaches between pediatric and adult care providers which have already been mentioned in the literature [8,22,37]: pediatric care delivered by multidisciplinary teams and individually oriented adult care; lack of multidisciplinary teams in adult care; and almost no attention paid to psychosocial support. The latter is largely missing in the transition to adult care, as almost all the participants mentioned it in the focus groups, even though the research on transition of care emphasizes the importance of psychosocial services in the transition of care [20]. This finding is all the more striking and makes the push for psychosocial support in the adult care vital, since the literature on quality of life in adult people with VSCs reveals that they experience a range of serious health issues, such as depression, anxiety, and suicidality [24,44,45], in comparison to the general population; feel stigma around their bodies and sexualities [46,47,48]; and age in poor psychosocial conditions [24]. Even though the rights of children did not feature prominently in the discussion, they were in the background as a position, because the participants highlighted the importance of child-oriented care and the time and resources dedicated to pediatric care. The lack of psychosocial support in this study is mirrored by previous studies on the service experience of young people with VSCs, which showed that psychosocial support is not provided even when it is asked for [41], and that young people with VSCs are not often referred for this support for psychosocial reasons [29]. The lack of psychosocial support leaves adults feeling abandoned and exacerbates stigma and shame [37].

There is yet another, more important duty for health care professionals to improve transition of care for young adults and older people with VSCs, and that is the history of medical treatment for people with VSC. The medical practices prior to the Chicago consensus statement had poor surgical outcomes [11] and deleterious effects on the mental health of people with VSCs [49], since it did not abide by bodily autonomy or truthful informed consent policies and seriously lacked psychosocial support [15]. The need and ethical imperative for transition of care is, therefore, more urgent for the older and aging population of people with VSCs who are in need of medical treatment either due to medical harm or simply because their health is deteriorating. They need psychosocial support to help them undo the secrecy, stigma, and shame which they were told to live with in the past.

### Limitations

The results of this study cannot be generalized, since the sample was not representative. The drop-out of some participants also significantly influenced the final sample and the material used for the analysis. The overrepresentation of pediatric health care professionals and lack of involvement of adult care providers in the sample privileges the authoritative voice of pediatric health care providers, even though the author aimed to highlight the representation of peer support groups. Furthermore, the selection of participants mainly from the 9th DSD symposium led to the sidelining of critical individuals and non-mainstream practices in the medical field from the study.

## 5. Conclusions

The transition of care process for people with VSCs has been introduced, but lacks adequate implementation. The main hurdles to transition of care are a lack of resources and inadequate psychosocial care. There are not enough adult specialists to provide care for adolescents and young adults, and the patients have issues navigating the health care system as they must often educate the adult care providers. Psychosocial care remains insufficiently provided, especially for older people with VSCs whose quality of life is below optimal due to problems with medical care and poor social adaptation. Multidisciplinary teams should receive additional resources and shift the care approach toward a psychosocial model to effectively address the ongoing needs of individuals with VSCs. Future studies should systematically examine the factors and barriers to transition of care based on larger samples, and should address ways to improve care for adults and older people with VSCs. Future studies should also explore how gender as a variable operates in the provision of transition of care.

## Figures and Tables

**Table 1 healthcare-12-00354-t001:** Coding tree.

Transition of care
Transition of care > Organization of transition of care
Transition of care > Organization of transition of care > Available
Transition of care > Organization of transition of care > Changes over time
Transition of care > Organization of transition of care > Document transfer
Transition of care > Organization of transition of care > Criteria for transition
Transition of care > Organization of transition of care > Adult specialists
Transition of care > Organization of transition of care > Adult specialists > Establishing contact with and informing the adult specialist
Transition of care > Organization of transition of care > Adult specialists > Education of the adult specialist
Transition of care > Organization of transition of care > Adult specialists > Education of the adult specialist > By pediatric specialist
Transition of care > Organization of transition of care > Adult specialists > Education of the adult specialist > By intersex people themselves
Transition of care > Organization of transition of care > Which specialties are involved
Transition of care > Psychosocial support for adults (lacking?)
Transition of care > Hurdles to implementation
Transition of care > Hurdles to implementation > Organization of medical specialties
Transition of care > Hurdles to implementation > Lack of related specialists/resources
Transition of care > Hurdles to implementation > Access to regional centers
Transition of care > Hurdles to implementation > Connection among MDTs and regional centers
Transition of care > Hurdles to implementation > Financial costs
Transition of care > Hurdles to implementation > Transfer of knowledge

## Data Availability

The data are available upon reasonable request.

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
