# Peer review of "Challenges in Transition of Care for People with Variations in Sex Characteristics in the European Context"

_healthcare, 2024, doi:10.3390/healthcare12030354_

Round 1

Reviewer 1 Report

Comments and Suggestions for Authors

This paper is very clear and well written. 

Line 188 you say that people with VSC are introduced to "the idea of" transitional care. Did you mean that the long term plan for changing from receipt of care from adult rather than paediatric providers, or the initiation of transition processes?

The results section was interesting, with the 5 headings/ thematic labels which may, if included, have made the introduction more interesting.

In the documentation section, the practice of using "special folders" is innovative practice, the extent of which should be mentioned i.e. one unit, one country etc.  Hand held notes is used in other areas of health care and this might be mentioned. 

While the resource-based barriers to transition are already known informally, the study does reveal  particularly interesting novel finding, that being the transfer of knowledge.  I have observed that paediatric services hold concern that adult providers will have insufficient knowledge with which to provide quality care for adults with VSC. I have not seen this written about ever. The author could make a little more of this important finding.

The limitations section could be softened to less undermine the value of the findings. I see the limitations as including self selection of participants with the most well developed services an the alack of involvement of adult providers who share equal responsibility for transitional care.

Finally, I find the abstract quite repetitive and not as exciting a  "shop window" for the paper as it could be.

Author Response

Reviewer 1

This paper is very clear and well written. 

Dear reviewer 1, thank you very much for taking the time to review this manuscript. Your comments were very insightful, and they helped me improve the quality of the paper. Please find the detailed responses below and the corresponding revisions/corrections highlighted/in track changes in the re-submitted files.

Line 188 you say that people with VSC are introduced to "the idea of" transitional care. Did you mean that the long term plan for changing from receipt of care from adult rather than paediatric providers, or the initiation of transition processes?

Response 1:

Dear reviewer 1, thank you for pointing out the lack of clarity in the text. What I refer to is the transition process. I have changed the mentioned part accordingly (line 216): 

As participants reported the transition process for adolescents, young adults, and parents is initiated early on.

The results section was interesting, with the 5 headings/ thematic labels which may, if included, have made the introduction more interesting.

Response 2:

Dear reviewer 1, thank you for pointing this out. I appreciate your suggestion, but I would prefer to keep the introduction and the results section as it is to emphasize the difference between the literature overview and the deductive outcome of the reflexive thematic analysis. I think that the current layout emphasizes my input and the themes well.

In the documentation section, the practice of using "special folders" is innovative practice, the extent of which should be mentioned i.e. one unit, one country etc.  Hand held notes is used in other areas of health care and this might be mentioned.

Response 3:

Dear reviewer 1, thank you for pointing this out. I followed you great suggestion to emphasize the special folders as innovative practice. The added parts read as (line 259):

Participants from Germany and Slovenia mentioned that children with VSC and their parents in most teams receive the copies of the diagnosis and provided treatment and they are also given a special folder in which they can note their questions, remarks and impressions received during consultations and collaborations with the health care providers in the teams and peer support groups.

(line 321):

Minority of teams not only introduced early transition process as pediatric health care professionals feel responsible to educate adult health care providers. What is more, the few pediatric teams even introduced an innovative practice in documentation as they provide special folders for parents to keep track of health care delivery and discussions with health care professionals. Special folders seem to be unique in the care for people with VSC as in other areas of health care handheld notes are used.

While the resource-based barriers to transition are already known informally, the study does reveal particularly interesting novel finding, that being the transfer of knowledge. I have observed that paediatric services hold concern that adult providers will have insufficient knowledge with which to provide quality care for adults with VSC. I have not seen this written about ever. The author could make a little more of this important finding.

Response 4:

Dear reviewer 1, thank you for highlighting the transfer of knowledge as an important and novel practice. I was not aware that it is that rare and therefore I really appreciate that you pointed it out. I have changed the text as follows (line 287):

On the other hand, pediatric care providers strive to fill in the gap in insufficient knowledge in adult care providers which is extremely rare practice.

The limitations section could be softened to less undermine the value of the findings. I see the limitations as including self selection of participants with the most well developed services an the alack of involvement of adult providers who share equal responsibility for transitional care.

Response 5:

Dear reviewer 1, thank for pointing this out. I followed your advice and revised the limitations sections. The changed text now reads as (line 409):

The overrepresentation of pediatric health care professionals and lack of involvement of adult care providers in the sample privileges authoritative voice of pediatric health care providers even though the author aimed to highlight the representation of peer support groups. What is more, the selection of participant mainly from the 9th DSD symposium led to sidelining of critical voices and non-mainstream practices in the medical field from the study.

Finally, I find the abstract quite repetitive and not as exciting a  "shop window" for the paper as it could be.

Response 6:

Dear reviewer 1, thank you for pointing this out. To the best of my abilities, I tried to revise the abstract to make it more interesting. It now reads as (line 7):

Abstract: Objective: People with variations of sex characteristics (VSC) have receiving inadequate care for many decades. The Chicago Consensus statement 2006 aimed to introduce improved comprehensive care which would include the transition of care from pediatric to adult services organized by multidisciplinary teams. Yet, the evidence for transitional care is scarce. The aim of this paper is to outline the delivery of transition of care for adolescents and young adults with VSC. Method: Seven focus groups were conducted with health care profes-sionals and peer support groups in care teams in Central, Northern and Western Europe. The data from the focus groups were examined using reflexive thematic analysis. Results: Even though the transi-tion of care has been implemented in the last two decades it remains inadequate. There are differ-ences among countries as the quality of care depends on available resources and variation of sex characteristics. Moreover, there are significant hurdles to adequate transition of care as there is lack of time and funding. Lack of adult care providers and psychosocial support often leaves young adults with VSC navigating the health care system alone. Conclusion: The outcome of the study shows that the transition of care is organized through the department of pediatric endocrinology. The quality of care varies due to resources and variation of sex characteristics. Lack of adult spe-cialists, especially psychosocial support, represents the biggest obstacle for young adults and adults to navigate the healthcare system and for the improvement in provision of healthcare to adults. There is risk od re-traumatization as adolescents and young adults must often repeat their medical history and educate to adult care providers who are insufficiently trained and knowl-edgeable.

Reviewer 2 Report

Comments and Suggestions for Authors

Dear Team

Thank you for allowing me the opportunity to review your work. 

I offer some questions to help clarify the work further.

Please explain collaborative care (I think the essence is that this is delivered via MDT due to there being an MDT?). The guidelines address equality and quality in care delivery; transition is included as an integral aspect of this. What the paper expresses is more transfer versus transition. What is the aim of the paper, considering the limited transition studies and evaluation, how did this inform the current study design and approach? The current paper suggests it is focused on finding ‘opinions’ from the MDT team and peer support groups for children, young adults and parents (this is three different groups – who is the population of interest here).

The study design is exploratory and qualitative, with only one reference to the approach, which neglects some key analysis that needs to be described and evidenced in the paper.

What is quality (within this paper) – what does it mean and what is the teams position on transition?

How were the questions for the FG developed and refined? Where are they?

What are the positions of the researchers and does this bring equity in gender and limit bias? What was the driving epistemology – is this work framed from a position of equity?

Who is the informant? The participant?

Recruitment was from an already invested group as they attended the meeting – how does this influence findings? How did recruitment happen outside of this?

The process pieces in the paper can be shifted to a figure or a table – to allow for a richer description of the study.

Where/when did the pilot take place, and how did this inform aspects of the study?

The analysis section could be supported with a clear figure as to who was involved at what stage and what happened in those dialogues.

There was an agreement of three themes, so what was neglected and why these three? Was the work inductive or somewhat deductive by the nature of the roles the researcher holds?

If quality and provision are critical aspects then these concepts need to be clearly outlined in the paper, as these can vary.

Team 5 is missing in the results – why?

The findings at times contradict – which is interesting as it highlights the complexity in this work. Also, there is language that situates gender and stance as authoritative (hence me asking about the position of the researchers).

It is important to highlight that transfer/transition can only really happen if there is someone to receive the patient, and the gaps in adult care are significant. What drives this politically and within funding and policy would be good to see in the discussion. The authors draw out the cycle of repeat and the need to be the educator – which is a way in which we re-traumatize populations, which can lead to disengagement and mistrust. 

The work is important in this small field, and there is rich learning from this work, it would be helpful to see this presented.

Comments on the Quality of English Language

none

Author Response

Dear Team

Thank you for allowing me the opportunity to review your work.

Dear reviewer 2, thank you very much for taking the time to review this manuscript. Your comments were very insightful and they helped me improve the quality of the paper. Please find the detailed responses below and the corresponding revisions/corrections highlighted/in track changes in the re-submitted files.

I offer some questions to help clarify the work further.

Comment 1: Please explain collaborative care (I think the essence is that this is delivered via MDT due to there being an MDT?). The guidelines address equality and quality in care delivery; transition is included as an integral aspect of this. What the paper expresses is more transfer versus transition. What is the aim of the paper, considering the limited transition studies and evaluation, how did this inform the current study design and approach? The current paper suggests it is focused on finding ‘opinions’ from the MDT team and peer support groups for children, young adults and parents (this is three different groups – who is the population of interest here).

Response 1:

Dear reviewer 2, Thank you for pointing this out. I agree with this comment. I took you advice and explained what collaborative care is and the added part now reads as (line 32):

Collaborative care is a multiprofessional approach to patient care based on clinical evi-dence and standardized tools to provide care tailored for an individual accompanied by follow-ups and communication among health care professionals [3].

The aim of the study was a bit unclearly defined and the population of interest which were health care professionals and peer support groups were ill-defined. Therefore, I have clarified the aim of the study, and it now reads as (line 91):

As only a few studies explored the challenges in transition of care for adolescents and young adults with VSC revealing issues related to lack of resources and differences in care approaches this study took explorative design and approach based on reflexive thematic [29,30] analysis that would enable the author more reflexive and subjective outlook to outline the scope of issues in the field. The aim of the study is to map out the delivery of transition of care for adolescents and young adults with VSC based on focus groups the with peer support groups and members of multidisciplinary teams working in care for children with VSC.

The study design is exploratory and qualitative, with only one reference to the approach, which neglects some key analysis that needs to be described and evidenced in the paper.

Response 2

Dear reviewer 2, Thank you for pointing this out. I agree that there is only one reference to the approach which led to neglecting some key aspects in the study. I tried to add more references to the approach and explain more in-depth the relevance of the approach in the following way:

(line 91)

As only a few studies explored the challenges in transition of care for adolescents and young adults with VSC revealing issues related to lack of resources and differences in care approaches this study took explorative design and approach based on reflexive thematic [29,30] analysis that would enable the author more reflexive and subjective outlook to outline the scope of issues in the field.

(line 104):

Reflexive thematic analysis approach is [30] flexible enough to enable the author to use their creative and critical thinking skills to draw out the underlying themes from the focus groups.

What is quality (within this paper) – what does it mean and what is the teams position on transition?

Response 3:

Dear reviewer 2, Thank you for pointing this out. None of the objectives of the study explicitly addressed the quality of care as the study rather inquired about the provision of care on a more descriptive level rather than on the evaluative one. Neither the questions nor the informants’ answers explicitly tacked the issue of quality of transition of care. As the reflexive thematic approach gives the analyst a lot of space to deductively examine the data and formulate the themes this led me to use the word quality of transition of care. I took your advice and included the definition of quality of care based on the EU and WHO standards in the introduction because it gives better insight into the quality of care. The added part in the introduction now reads as (line 36):

Quality of care consists of effective, safe and patient-centered care as well as accessibility, efficiency and equity of care [5,6].

The added part in the discussion now reads as:

(line 329):

The results of the study suggest that the quality of transition of care is not effective, safe and patient centered as care services suffer from lack of adequately skilled health care professionals and poor outreach of the adolescents and young adults with VSC.

How were the questions for the FG developed and refined? Where are they?

Response 4

Dear reviewer 2,

Thank you for pointing this out. The interview guide for the FG were developed based on the literature overview on the collaboration among health care professionals, peer support groups, children with VSC and their parents. The interview guide was then discussed and revised with the supervisors and a fellow PhD candidate in two rounds. As you expressed the wish to see the interview questions, I added them as the supplement 1 to the paper. The added part on the development of focus groups now reads as following in the text (line 142):

The author drafted the first version of the interview guide based on the scoping review [36] and additional literature. Then the author discussed the interview guide with su-pervisors and two PhD Candidates on the INIA (Intersex – New Interdisciplinary Ap-proaches) project. Thereafter the author added the quotes as suggested by the supervisors to use them in case the participants struggle to find the answers. After the second round of consultation with supervisors and the fellow PhD candidates the author changed wording in some place.

What are the positions of the researchers and does this bring equity in gender and limit bias? What was the driving epistemology – is this work framed from a position of equity?

Response 5

Dear reviewer 2,

Thank you for pointing this out. I agree that the positionality of the research(er) is not clearly defined. I tried to provide more information about it. The driving epistemology in this study was based on concepts of pediatric shared decision making and multi/interdisciplinarity and children’s rights. I have accordingly modified the text which now reads as following (line 107):

Study design was informed by the concepts of pediatric shared-decision making [33], multi/interdisciplinarity [34] and children’s rights in health care [35].

I hope I understood your critique about the equity in gender correctly and that the following answer will be in alignment with it. The gender equality implicitly informed the research, but it did not feature prominently for two reasons. Firstly, the more explicitly gender approach would require a separate analysis of how operates in the transition of care for adolescents and young adults with VSC which is out of the scope of the study. Secondly, the author did not ask participants about their gender (race, ethnicity or any other social variable other than the medical profession) and therefore any analysis of how gender operates in the transition of care for adolescents and young adults with VSC would be based on the assumptions of the author and therefore not empirically examined. However, I agree that this is an important research topic that is worth exploring and therefor I decided to add it to the conclusion section which now reads as (line 427):

Future studies should also explore how gender as variable operates in the provision of transition of care.

Who is the informant? The participant?

Response 6

Dear reviewer 2,

Thank for pointing out the unclearly defined interchangeable use of the words informant and participant. I have accordingly corrected the use in the text and decided to use participant only (line 113):

Selection of Participants

The inclusion criteria for the participation in the study were: participants had to be health care professionals working in a European multidisciplinary team providing care for children with VSC and their parents, or members of peer support groups involved in any way in collaboration with multidisciplinary teams.

Recruitment was from an already invested group as they attended the meeting – how does this influence findings? How did recruitment happen outside of this?

Response 7

Dear reviewer 2,

thank you for pointing this out. The selection of participant that happened at the 9th DSD Symposium led to inclusion of mainstream voices in the study. More critical or invisible voices were therefore obscured, and variety practices excluded from study. Outside the meeting the participants were selected through the convenience sampling. I have accordingly revised the text and now it reads as (line 410):

The overrepresentation of pediatric health care professionals and lack of involvement of adult care providers in the sample privileges authoritative voice of pediatric health care providers even though the author aimed to highlight the representation of peer support groups. What is more, the selection of participant mainly from the 9th DSD symposium led to sidelining of critical voices and non-mainstream practices in the medical field from the study.

(line 126):

The minority of team leaders and the team members as well the members of the peer support groups were selected through convenience sampling, that is separately from the 9th DSD Symposium.

The process pieces in the paper can be shifted to a figure or a table – to allow for a richer description of the study.

Response 8,

Dear reviewer 2,

Thank you for pointing this out, but I am not sure I understand what you are referring to. If I understood you correctly you suggest removing the pieces of information on the process from the main text and put them in a table or a figure? This would mean that I would have to change the text quite substantially and it would disrupt the consistency of the themes.

Where/when did the pilot take place, and how did this inform aspects of the study?

Response 9,

Dear reviewer 2,

Thank you for pointing this out. I have added the information about the time and place of the pilot, but not in details as this would very easily lead to identification of participants and breach the confidentiality clause in the informed consent. Moreover, the pilot helped me to make the interview guide shorter and be more assertive as the interviewer. The added part now reads as (line 137):

The pilot was conducted with a team from Switzerland in May 2022. The pilot helped the author to shorten the interview guide and it helped the author to be more assertive in managing the time and group dynamics during the focus groups.

The analysis section could be supported with a clear figure as to who was involved at what stage and what happened in those dialogues.

Response 10,

Dear reviewer 2,

thank you for pointing this out. I am afraid I cannot adequately meet you request as this was out of the scope of the study. Reflexive thematic analysis is not best suited to examine the dynamics in focus groups. What is more including the dynamic of the focus groups would in my opinion would lead to complications in analysis and would create confusion about aims of the paper.

There was an agreement of three themes, so what was neglected and why these three? Was the work inductive or somewhat deductive by the nature of the roles the researcher holds?

Response 11,

Dear reviewer 2,

thank you for pointing this out. The work was mainly deductive which led the author to select the three themes as there were in my opinion the most common in all the focus groups. The themes like group dynamic and focus on biomedical knowledge were not included, because in my opinion the data was not rich enough to generate the common thread. The corrected part now reads as (line 193):

These themes featured most prominently in all the focus groups and were thus selected in the analysis. Themes like group dynamic and focus on biomedical knowledge were to be found only in some groups and the author could not find a common thread in all the focus groups.

If quality and provision are critical aspects then these concepts need to be clearly outlined in the paper, as these can vary.

Response 12,

Dear reviewer 2,

thank you for pointing this out. The participants were not explicitly asked about the quality of care was, but as the reflexive thematic analysis allows author to generate themes it seemed suitable to me to mention the concept as it clarifies the issues in transition of care. The added part now reads as (line 36):

Quality of care consists of effective, safe and patient-centered care as well as accessibility, efficiency and equity of care [5,6].

Team 5 is missing in the results – why?

Response 13,

Dear reviewer 2,

thank you for pointing this out. I agree that the team 5 is somewhat underrepresented as the data was not as rich as in other focus groups, but the team 5 is not missing completely. Accordingly, I emphasized the mentioned part and the team in the paragraph 25 (line 281):

Although the number of specialists, of adult specialists interested in Turner DSD-pathology is rare, very rare. Adult specialists have rarely interest in these conditions, there are so many other pathologies that are much more frequent, that you may be grateful if you find somebody who is interested and has some experience and knows how to take care of adult patients. (a member of a peer support group, FG 5)

The findings at times contradict – which is interesting as it highlights the complexity in this work. Also, there is language that situates gender and stance as authoritative (hence me asking about the position of the researchers).

Response 14,

Dear reviewer 2,

thank you for pointing this out. Although the research indirectly touched on gender equality, it wasn't emphasized for two reasons: delving into explicit gender analysis in transition of care is beyond the study's scope, and participant data on gender wasn't collected, making any gender-related analysis non-feasible. The overrepresentation of health care professionals in the sample led to authoritative stance of health care professionals on the topic. Thereby experiences and opinions of peer support groups were sidelined even though the author aimed to highlight them. I addressed the issue of overrepresentation in the limitations section as follows (line 410):

The overrepresentation of pediatric health care professionals and lack of involvement of adult care providers in the sample privileges authoritative voice of pediatric health care providers even though the author aimed to highlight the representation of peer support groups. What is more, the selection of participant mainly from the 9th DSD symposium led to sidelining of critical voices and non-mainstream practices in the medical field from the study.

It is important to highlight that transfer/transition can only really happen if there is someone to receive the patient, and the gaps in adult care are significant. What drives this politically and within funding and policy would be good to see in the discussion. The authors draw out the cycle of repeat and the need to be the educator – which is a way in which we re-traumatize populations, which can lead to disengagement and mistrust.

Response 15,

Dear reviewer 2,

thank you for pointing this out. I agree that transfer and/or transition can take place if there is someone to receive the patient. Thank you for encouraging me to explore the political and economic reasons for inadequate transition of care. Accordingly, I added the part (line 339):

Our study went further and showed that transition of care often cannot happen as there are often simply no adult care providers for adolescents, young adults, and older people with VSC. The lack of adult providers for adolescents and young adults with VSC corresponds to lack of access to health care professionals in Europe [39]. For example, in Germany the doctors are unequally represented throughout the regions, whereas in the United Kingdom not only are doctors more concentrated in some regions, but there is an overall lack of health care professionals. The uneven representation of health care professionals is further complicated by the fact that medical specialties like internal medicine, psychiatry, pediatrics, pathology and neurology receive more funding than family medicine, gynecology and plastic surgery [40]. Additionally, variations of sex characteristics are seen as non-viable option for the future by some health care professionals and parents [41]. The growing child is envisioned to dispense with a given variation and fit into the “cis-gendered future”[17]. The discourse of cis-gendered future informs the medical practice as it signals that adult people with VSC had their variation removed or made invisible and the care for it is no longer needed.

(line 360)

The need to repeat the medical history to inadequately educated and trained adult professionals may re-traumatize the adolescents and adult people with VSC.

The work is important in this small field, and there is rich learning from this work, it would be helpful to see this presented.

Response 16,

Dear reviewer 2,

Thank you for your encouraging words.

Reviewer 3 Report

Comments and Suggestions for Authors

This is an excellent paper I was absolutely thrilled to read it. There is a dearth of literature on the transition from pediatric to adult care in this area and the authors respond to this magnificently. 

My only criticism is that the conclusion ends up being slightly tentative. Whilst we have seen (huge?) overhauls of MDT's and care in childhood services we have seen nowhere near the same level of investment in adult care. I think the conclusion can be slightly bolder in calling for the urgent redirection of resources towards developing MDT's capable of responding to the long-term needs of people with VSC. These have to be developed through a psychosocial care model - if only to respond to the traumas that many of these people will have undergone at the hands of medicine. This is discussed in the final chapter of Garland and Travis' new book!

Otherwise I think is a well worked intervention in an area that has a distinct need for data on medical professionals. 

Comments on the Quality of English Language

Quality was good but some minor errors. A thorough proof reader will pick them up.

Author Response

Dear reviewer 3, thank you very much for taking the time to review this manuscript. Your enthusiasm  assured me that my work is valuable. Please find the respond to your comment below.

My only criticism is that the conclusion ends up being slightly tentative. Whilst we have seen (huge?) overhauls of MDT's and care in childhood services we have seen nowhere near the same level of investment in adult care. I think the conclusion can be slightly bolder in calling for the urgent redirection of resources towards developing MDT's capable of responding to the long-term needs of people with VSC. These have to be developed through a psychosocial care model - if only to respond to the traumas that many of these people will have undergone at the hands of medicine. This is discussed in the final chapter of Garland and Travis' new book!

Response 1,

Dear reviewer 3,

Thank you very much for emboldening me to write bolder conclusion. I agree that care approach should shift toward psychosocial model suggested by Garland and Travis. I recently started reading their book and find the last three chapters truly informative. I added the part and now reads as (line 423):

Multidisciplinary teams should receive additional resources and shift the care approach toward psychosocial model to effectively address the ongoing needs of individuals with VSC.

Otherwise I think is a well worked intervention in an area that has a distinct need for data on medical professionals.

Round 2

Reviewer 2 Report

Comments and Suggestions for Authors

please see attached 

Comments on the Quality of English Language

needs editing

Author Response

Review

Thank you for answering the questions.

Dear reviewer 2, thank you very much for taking the time again to review this manuscript. Your comments were very insightful, and they helped me improve the quality of the paper. Please find the detailed responses below and the corresponding revisions/corrections highlighted/in track changes in the re-submitted files.

I offer some thoughts and clarification.

Suggestion: Page 2 line 47: The consensus statement acknowledges the historic deficiency in the transition of care [18], briefly mentioning that one of the responsibilities of multidisciplinary teams is coordinating the transition of care from paediatric to adult settings.

Response 1

Dear reviewer 2, Thank you for your suggestion. I took your suggestion and incorporated in the text. It now reads as (page 2, line 47):

The consensus statement acknowledges the historic deficiency in the transition of care [18], briefly mentioning that one of the responsibilities of multidisciplinary teams is coordinating the transition of care from paediatric to adult settings.

Within this section there is some repeat that could be removed with editing – the above is one example. Further, there is some slippage in statements that do not reflect the language in the rest of the paper – for example, “lot of room for improvement.” – what does this mean? Could it be merged with the following sentences, perhaps;

Although there is increasing awareness (or agreement- I was unsure as to which) among professionals regarding the requirements for transitioning care, the evidence indicates that there is still significant opportunity for enhancement [21]. Adults with VSC experience health problems that have been either created or worsened by medical interventions [22]. The research has identified several obstacles to the successful transition of care, for example, insufficient training in adolescent health care, potential diffusion of responsibility among paediatric and adult care providers, the complexity of health care conditions, inadequate communication between paediatric and adult health care providers, the need for meaningful inclusion of adolescents and young adults in the transition process, lack of planning and coordination of services, and insufficient evidence to guide the development of transition of care [21,23–25].

Only a limited number of studies have investigated the difficulties in transitioning care for adolescents and young adults with VSC. These studies have identified problems such as limited resources and variations in care approaches [Refs]. The current study employed an explorative design to address this gap and utilized reflexive thematic analysis [31,32].  

Response 2

Dear reviewer 2, Thank you for this great suggestion. I agree that “room for improvement” is not the most suitable wording. I took your suggestion and changed the text as you suggested. It now reads as (page 2, line 47 – line 90):

The consensus statement acknowledges the historic deficiency in the transition of care [18] briefly mentioning that one of the responsibilities of multidisciplinary teams is co-ordinating the transition of care from paediatric to adult settings [4]. The consensus update added that people with VSC share challenges in accessing adult care with other people with chronic conditions [10]. Moreover, the consensus update also highlighted the reasons for lack of appropriate transition of care: adolescents and young adults avoid care pro-viders due to poor outcomes of the past medical treatments, non-disclosure of diagnosis and medical treatment in the past and anxiety related to sexual and romantic relations [10]. In the research on adjustment from pediatric to adult care transition of care is defined as a process that includes services aimed at planned and coordinated shift between pe-diatric and adult care for young people with chronic conditions[19–21].

The few recommendations on transition of care for adolescents and young adults with VSC that were published in the years between the consensus and its update in 2016 stipulated that the transition of care is a multiple-year process that requires a plan and review for each person with VSC according to their variation and needs, assessment of readiness for the transition and assistance in guiding and education a person with VSC and their family [18,26–28]. Furthermore, the recommendations highlighted the im-portance of continuous collaboration between pediatric multidisciplinary team, parents, peer support groups and adult providers who should be involved in the care as early as possible, that adult providers should have good communication skills and be well-versed in issues surrounding care for people with VSC. These recommendations also pointed out that transition is a process that involves a shift in care approach and it should start at the age of 12-13 year and be finished around the age of 25, whilst the transfer is a step in which a patient stops seeing a pediatric health care professional and makes an appointment with adult care provider [18,27].

Although there is increasing awareness among professionals regarding the re-quirements for transitioning care, the evidence indicates that there is still significant opportunity for enhancement [21]. Adults with VSC experience health problems that have been either created or worsened by medical interventions [22]. The research has identified several obstacles to the successful transition of care, for example, insufficient training in adolescent health care, potential diffusion of responsibility among paediatric and adult care providers, the complexity of health care conditions, inadequate communication between paediatric and adult health care providers, the need for meaningful inclusion of adolescents and young adults in the transition process, lack of planning and coordination of services, and insufficient evidence to guide the development of transition of care [21,23–25]

Only a limited number of studies have investigated the difficulties in transitioning care for adolescents and young adults with VSC [22]. These studies have identified problems such as: a lack of trained providers in the adult care for people with VSC and their parents, a lack of priority and finances for multidisciplinary care in healthcare systems that VSC require, differences in care approaches between pediatric and adult care, a lack of psychosocial support and a ack of data the provision of transition of care [26,27,29]. The scarce evidence on the transition of care points for need to examine the poor implementation of transitional care [29,30].

At this point – there is an opportunity to be clear about how and where the researcher ‘fits’ as a reflexive approach has to consider this. If you are framing as deductive, latent, critical and relativist, constructionists as Braun and Clarke often tend to have these in combination – you can make your approach explicit within the methodology. This, then, is the gender piece) it's about the team, not the participants, as we, by virtue of our work and position, bring beliefs and biases).

Response 3

Dear reviewer 2, Thank you for pointing this out. I hope I understood your suggestion this time. The added part now reads as (page 2, line 93):

The author of present study is part of broader research project called INIA – Intersex New Interdisciplinary Approaches that aims to develop knowledge that will inform policy making and practice that supports the wellbeing and social and economic contributions of people with variations of sex characteristics. Participation in the project enabled the author to become familiar with the issues in care for people with VSC and the relevant stakeholders. The research question was addressed by constructionist, experiential, inductive, critical and latent framework.

I can see you have much more in the design section, its still feels like a list rather than a way of guiding the reader as to how and why you landed here and how you positioned the work etc. you have meaning as a group and then focus on the author.

Response 4

Dear reviewer 2, Thank you for pointing this out. I agree with you that the text at times feels like a list. I took your advice and put the study in context and more clearly positioned myself as author. The added parts now read as:

(page 2, line 91):

This study took explorative design and approach based on reflexive thematic analysis [31,32] that would enable the author more reflexive and subjective outlook to outline the scope of issues in the field. The author of present study is part of broader research project called INIA – Intersex New Interdisciplinary Approaches that aims to develop knowledge that will inform policy making and practice that supports the wellbeing and social and economic contributions of people with variations of sex characteristics. Participation in the project enabled the author to become familiar with the issues in care for people with VSC and the relevant stakeholders. The research question was addressed by constructionist, experiential, inductive, critical and latent framework.

(page 3, line 148):

Then the author discussed the interview guide with supervisors (a medical doctor and a philosopher – none of them a person with VSC) and two PhD Candidates on the INIA project, both people with VSC. The variety of professional backgrounds and characteristics in the discussions helped the author to approach the research with respect for people with VSC while staying critical and attuned to the complexity of the care for people with VSC. The collaboration with supervisors and two fellow INIA PhD candidates enabled the author to enlarge the scope of interview guide, deepen the analysis of data and enrich the development of themes. Thereafter the author added the quotes as suggested by the supervisors to use them in case the participants struggle to engage in conversation. After the second round of consultation with supervisors and the fellow PhD candidates the author changed wording in some places.

The rights of the child – great, do you feel this is in the background as a position, it is maybe a little hidden, can you make it explicit?

Response 5

Dear reviewer 2, Thank you for pointing this out. The rights of the child did not feature prominently in the discussion on the transition of care as they were in the foreground when the pediatric care was discussed during the conversations. I did feel this was in the background as a position. I took your suggestion and added the part in the text and now it reads as (page 9, line 399):

Even though the rights of child did not feature prominently in the discussion they were in the background as a position because the participants highlighted the importance of child-oriented care and the time and resources dedicated to the pediatric care.

Pg 3 line 146 Can I ask - struggle to find the answers. The FG questions are prompts to help guide the discussion and conversation – how can there be ‘answers’?

Response 6

Dear reviewer 2, Thank you for pointing this out. I agree that the phrasing is a bit clumsy. Therefore I changed the text and now reads as follows (page 3, line 155):

Thereafter the author added the quotes as suggested by the supervisors to use them in case the participants struggle to engage in conversation.

Are the supervisors academic or working in the field?

Response 7

Dear reviewer 2, Thank you for pointing this out. One supervisor is academic and works in the field. The other is just academic. I added the suggestion in the text and now reads as (page 3, line 129):

their first supervisor who is an academic and physician in the field. The second supervisor is only an academic.

Discussion – pg 9 line 394-5 What is more there is an unaddressed need in the adult care for those adult people with VSC who seek gender transition due to gender misassignment early in life

There is nothing in your work to suggest this, and while it's interesting, it's very small numbers, and I caution you not to connect the two in this paper.

Response 8

Dear reviewer 2, Thank you for pointing this out. I took your suggestion and removed that part from the text.

Overall, the paper is interesting and its will add value to the field, the methods/design still need some clarity and the paper needs to be edited to remove repeats and to tighten flow and grammar. Adding complex issues such as gender transition to this work is overly complicated and there is nothing in the data reported to support this. The rights of the child and individual can be strengthened (briefly in the background) – are they evident in the findings and then would have a clear position in the discussion.

The addition of the legacy and the impact on the now adult population is well done – it’s so often neglected, and your data highlights the challenges for this group.

Response 9

Dear reviewer 2, Thank you for pointing this out. The introduction has been changed and corrected to improve the flow of the text. The transition and the rights of children have been addressed accordingly to your suggestions and I hope the paper has been sufficiently improved.

This paper is not just about CAH – table 1 identifies the spectrum Sanders, C.; Carter, B. A Qualitative Study of Communication between Young Women with Disorders of Sex Development 534 and Health Professionals. Adv. Nurs. 2015, 2015, e653624, doi:10.1155/2015/653624.

Response 10

Dear reviewer 2, Thank you for pointing this out. I corrected the mistake in the text. The changed part now reads as (page 9, line 379):

young women with VSC

Have you reviewed this paper?

Grimstad F , Kremen J , Streed CG , et al . The health care of adults with differences in sex development or intersex traits is changing: time to prepare clinicians and health systems. LGBT Health 2021;8:439–43.doi:10.1089/lgbt.2021.0018

Response 11

Dear reviewer 2, Thank you for bringing this paper to my attention. I am afraid I have not reviewed this paper as I do not have the access to it.

Also – I note this paper has a great deal of overalp with Gramc, M. Challenges in Transition of Care for People With Variations of Sex Characteristics in the European Context. Preprints 2023, 2023110880. https://doi.org/10.20944/preprints202311.0880.v1 - Can I ask how is it possible to print this twice?

Response 12

Dear reviewer 2, as I have been told by the editor, this has already been explained to you by the editor.